# Method for 3D City Building Continuous Transformation Based on an Improved LOD Topological Data Structure

**Siyi Li** [1,2]**, Wenjing Li** [1,2,*]**, Zhiyong Lin** [3] **and Shengjie Yi** [1]

1   College of Resource and Environmental Engineering, Wuhan University of Science and Technology, Wuhan 30081, China; lisiyi@wust.edu.cn (S.L.); yishengjie@wust.edu.cn (S.Y.)
2   Key Laboratory of Rock Mechanics in Hydraulic Structural Engineering, Ministry of Education, Wuhan 430081, China
3   School of Remote Sensing and Information Engineering, Wuhan University, Wuhan 430079, China; zhylin@whu.edu.cn
*   Correspondence: liwenjing@wust.edu.cn

**Abstract:** A 3D city model is an intuitive tool that is used to describe cities. Currently, level-of-detail (LOD) technology is used to meet different visual demands for 3D city models by weighting the rendering efficiency against the details of the model. However, when the visual demands change, the "popping" phenomenon appears when making transformations between different LOD models. We optimized this popping phenomenon by improving the data structure that focuses on 3D city building models and combined it with the facet shift algorithm based on minimal features. Unlike generating finite LOD models in advance, the proposed continuous LOD topology data structure is able to store the changes between different LOD models. By reasonably using the change information, continuous LOD transformation becomes possible. The experimental results showed that the continuous LOD transformation based on the proposed data structure worked well, and the improved data structure also performed well in memory occupation.

**Keywords:** topological data structure; 3D city buildings; level-of-detail; continuous transformation

## 1. Introduction

With the rise of data science, the urban informatization wave has had an enormous impact on various trades and occupations, and the "Smart City" has globally become a new theory and practice for next-generation urban development [1]. The Smart City is the space of virtual and real integration formed by the organic integration of digital and physical cities through the Internet. This space is also called the cyber–physical space [2], and the key to its development lies in an accurate understanding of geographic information. Three-dimensional (3D) city models can present geographic information in a more visually appealing manner [3] that can provide real, expressive models in urban planning and related areas [4]. The rational use of 3D city models will help elevate the level of city planning and provide references for actual construction processes, which also plays an important role in the rational allocation of construction costs and resources. 3D city models are considered to be the application foundation of urban planning thematic information visualization [5–9], which has become a research hot spot in the Smart City field.

However, in the application process of 3D cities, the enormous pressure on computer processing systems because of the tremendous amount of 3D building data thwarts the practical application of 3D city models to some extent. Therefore, rendering 3D building models with different levels of detail (LODs) on the premise of maintaining spatial geometric accuracy and human vision preference will

help to comprehensively understand buildings [10] and reduce the data pressure in the application process of 3D city models.

Aiming at the tension between the increasingly high requirement of realistic representations of complex scenes, the limited system hardware and software performance, and the cost of manpower and material resources in the widespread application process of 3D city models, LOD technology [11] is one of the most commonly used methods to solve this contradiction [12]. It is a widely used technique to solve the rendering problem for real-time 3D visualization, as rendering a massive amount of 3D geometric data requires considerable computer resources, sometimes far beyond the capacity of a graphic device. The LOD technique uses different models to represent an object, and the models have various degrees of details. According to the visual demands, computer resources can be allocated for rendering models; if the object is not particularly important, then models at a less-detailed level are chosen to obtain efficient rendering operation. By this means, the efficiency of rendering could be optimized and the visual effects could be closer to reality.

The LOD technique can be divided into discrete and continuous LODs [12]. The discrete LOD technique pregenerates and stores several models in different levels of detail. It omits the processing time of real-time simplified models; however, it occupies more storage space and mutates the visual effects while switching from one level of detail to another. Research efforts have yielded substantial results regarding the discrete LOD technique, from algorithms to specifications [13–16]. The continuous LOD technique dynamically generates models in different levels of detail according to the distance of the viewpoint and aims to transform models without visual catastrophes. The method by which to create the fast, real-time generation of continuous LODs is currently an important research issue. In research on 3DCM, great progress has been made on the technology of large-scale terrain continuous LODs based on real data. Many continuous LOD algorithms based on regular grids have realized the real-time display of large-scale terrain [17–21]. However, the research of continuous LOD technology for buildings is still in its infancy.

Currently, the progress of LOD algorithms for 3D building models is mainly embodied in the field of geometric simplification. Forberg [22] simplified 3D building features based on mathematical morphology. Baig [23], Ge [24], and Yan [25] simplified the characteristic surfaces (footprints and elevation of the building) of building models by limiting the number of edges, curves, and angles in the characteristic plane, thus simplifying the 3D building. The basic idea of this kind of method is to reduce the 3D model into a 2D surface and then simplify the 2D surface, which means only one aspect of the building is considered, making the overall expression effect of the 3D building difficult to guarantee. Regarding building simplification on a 3D level, Fan [26] and Xie [27] comprehensively considered the characteristics of the walls and roofs of buildings, independently simplifying the walls and roofs in the process of simplification and then connecting the simplified walls and roofs to complete the simplification of 3D buildings. Xie [10] studied simplification and aggregation methods of 3D buildings, and on this basis, realized the multiscale visualization of a 3D city.

However, current 3D building LOD algorithms are concerned with generating 3D building models of each LOD. Most studies pay little attention to the transformation mode and the improvement of the transformation effect between the 3D building models of each LOD. At present, the main method of continuous transformation of 3D building models is "fading", which is based on discrete LOD technology. It makes use of the transparent rendering effect of graphics hardware and realizes the transition of models at different levels of detail by setting transparent attributes of certain objects at different stages. However, this method must draw multiple models at different levels of detail at the same time, which occupies more system resources. Further, when the models transfer, the spatial location of some vertex changes to another position, thus causing visual incoherence, or the so-called "popping" phenomenon [28]. In order to improve the visualization effect of 3D scenes [29–31] and reduce the impact of visual mutation caused by the transformation of a model between different levels of detail on the visual effect, thus optimizing memory usage in the process of LOD model

transformation, it is necessary to study the detail-level transformation method of building models combined with a building LOD simplification algorithm.

In this paper, a data structure based on continuous LOD topology is proposed, which synthetically considers the general method of simplification of the 3D building LOD. The data structure constrains the edges and surfaces of 3D building triangular mesh by setting the corresponding relationship between the geometric data of the 3D building model in different LODs. According to the different stages of transformation, the usage state of the corresponding LOD geometric data is defined, thus realizing the continuous LOD transformation of a 3D building model.

## 2. Continuous LOD Data Structures for 3D Building Models

### 2.1. Essential Features of 3D Building Models

For each independent urban 3D building model, the structural information includes roofs, walls, footprints, windows, and so on. This structural information determines the shape and complexity of the 3D building model, among which the roof creates the biggest difference. Apart from some special buildings, the roofs of conventional buildings can be divided into several main forms, such as flat, spire, and sloped roofs.

According to whether the whole wall is vertical to the ground, buildings can be divided into two types: ordinary and special. Buildings with walls that are vertical to the ground are called ordinary buildings; very few buildings with nonvertical walls (mostly representative buildings in cities) are called special buildings. Compared with ordinary buildings, the structural information of special buildings is more complex and diverse; so, special methods are needed to synthesize them according to their shapes, which makes it difficult to process models in batches. However, for ordinary buildings, the overall shape of the building is always determined by the roof and the footprint, while the details of the building are determined by the structural information, such as windows and doors. Therefore, the batch processing of ordinary buildings is feasible as long as the processing method is reasonable. So, from the perspective of 3D building generalization, on the premise of guaranteeing the overall shape of the building, the simplification and restoration of 3D buildings need to be carried out from two aspects: roof and wall processing.

The walls of ordinary buildings can be roughly divided into three categories (Figure 1): flat walls, walls with uneven parts on one side, and walls with uneven parts on both sides. The surface constituting the wall is called the characteristic surface of the wall. The characteristics of the wall are related to the position and shape of the characteristic surface.

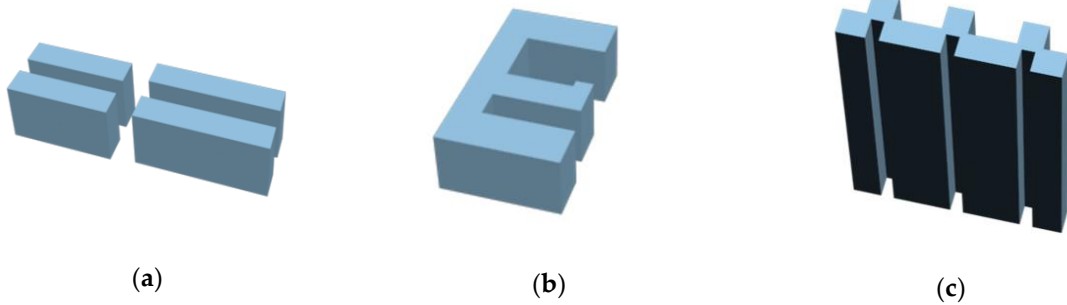

| (**a**) | (**b**) | (**c**) |

**Figure 1.** Wall types of ordinary urban buildings (in 3D): (**a**) flat walls, (**b**) walls with uneven parts on one side, and (**c**) walls with uneven parts on both sides.

The roof styles of urban buildings can be roughly divided into three categories (Figure 2): flat, spire, and sloped roofs.

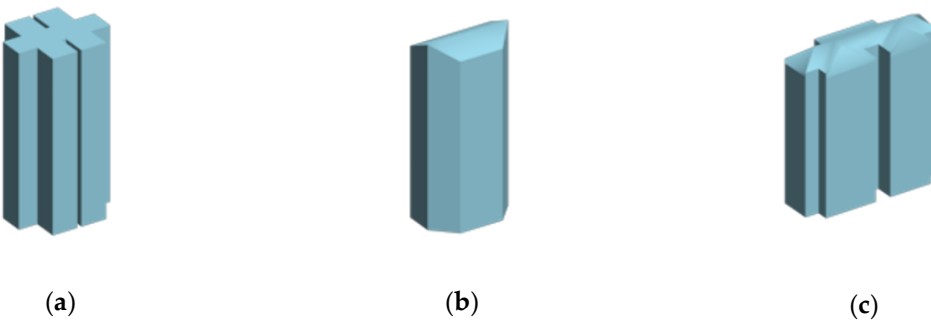

**Figure 2.** Roof types of ordinary urban buildings: (**a**) flat, (**b**) spire, and (**c**) sloped roofs.

The LOD simplification of 3D building models is different from other 3D models, in that it must keep the normal vector of the building surface unchanged (the wall is always vertical and the roof slope is fixed) and keep the basic shape of the 3D building model (the position and size of the characteristic surface) unchanged. Therefore, the key to simplifying the 3D building model is to deal with the details of the wall/roof while keeping the basic shape and location characteristics of the two main components (wall and roof) fixed.

### 2.2. Representation Method for 3D Building Model

In a 3D scene, the most commonly used method of model representation is boundary representation (BR). The core idea of BR is to represent the geometric shape and spatial position of a solid model with a vertex, edge, and triangle, which belongs to the representation method of solid boundary description. In BR, in order to operate all entity elements and other related parameters of the geometric model conveniently, the geometric features (position and direction) and topological structure of the entity model are preserved in detail, which has the advantage of representing the unique determination of the model.

2.2.1. Representation of Triangular Mesh

Geometric information includes the number of vertices, the coordinates of each vertex, the number of triangles, and the indexes of three vertices on each triangle. The simplified object of 3D architecture is mainly the geometric information of the model. In the triangular mesh model, each vertex belongs to two edges or more triangles, each edge connects two vertices (also belonging to two triangles), and the triangles are connected with each other.

The vertex table pointer method introduces a vertex table pointer when defining a polygon. When using this method, all vertices in the mesh model are stored only once in memory, which solves the shortcoming of repeated vertex storage in vertex direct representation. The vertex sequence V is shown in Figure 3, and then an index table is defined, which points to the vertex sequence V. In this figure, $P_1$ = (1, 2, 4), and $P_2$ = (4, 2, 3).

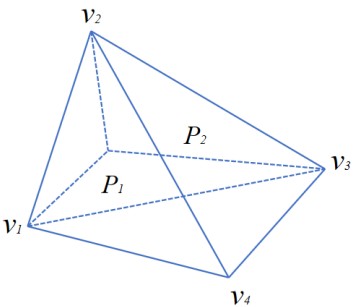

**Figure 3.** Vertex table pointer representation.

### 2.2.2. Orientation of Triangular Mesh

For any surface feature **P** in the 3D model, let the vertex be $(v_i, v_j, v_k)$. When the vertex of **P** is clockwise, the direction of **P** is defined as negative; otherwise, the direction of **P** is defined as positive. In order to express the direction of a triangle with a formula, let $e_{i,j}$ be the normal vector from $v_i$ to $v_j$, $e_{j,k}$ be the normal vector from $v_j$ to $v_k$, and define a vector **r** = (0,0,1). The direction of the triangle **P** is defined as **ORP** $(v_i, v_j, v_k)$, then

$$ORP(v_i, v_j, v_k) = \begin{cases} 1 & (e_{i,j} \times e_{j,k} \cdot r > 0) \\ 0 & (e_{i,j} \times e_{j,k} \cdot r = 0) \\ -1 & (e_{i,j} \times e_{j,k} \cdot r < 0) \end{cases} . \tag{1}$$

When the orientation of a triangle has **ORP** $(v_i, v_j, v_k) = 0$, it means the vertex of this triangle is on a straight line; it also means this kind of triangle will not be operated in 3D model scale transformation. When **ORP** $(v_i, v_j, v_k) \neq 0$, the vertices of the triangle are not collinear. If the triangle is located on a building characteristic surface, the triangle will participate in the translation operation. If the triangle is located on a noncharacteristic surface, the triangle itself will not take part in the translation operation, but there must be collinear points between the nonspecial fronts and feature surfaces, while these collinear points, as the constituent vertices of the feature surfaces, will still take part in translation operations.

### 2.3. Detail Transformation of 3D Building Model

Most urban buildings have parallel structures, and the characteristics mainly include two different forms: bulge and depression. In this study, the facet shift algorithm based on minimal features [32] was used to simplify the structure of the bulge or depression of a building with 3DS as the data output format. The model in 3DS format stores the geometric information of the model, including the number of vertices, the coordinates of each vertex, the number of triangles, and the index of three vertices on each triangle. The basic principle of the facet shift algorithm is that for the detailed model *M*, the feature with low importance (which was measured by the area proportion in this work) is moved to the feature with high importance each time, and the resolution is gradually reduced. Finally, the simplified 3D building model $M_0$ and a series of detailed record information are obtained. By reinserting nodes and triangles into the mesh, the model with its original resolution can be recovered.

The simplified process from *M* to $M_0$ is to delete a characteristic surface **P₁** from the mesh and project the predeleted characteristic surface **P₁** onto the prereserved characteristic surface **P₂** to form new nodes $V_a'$, $V_b'$, $V_c'$, and $V_d'$. The two noncharacteristic surfaces **NP₁** and **NP₂** that are adjacent to the characteristic surface are also deleted (as shown in Figure 4). As the surface translation operation proceeds, vertices and triangles in the 3D building model are gradually reduced; thus, the detail of the model is gradually reduced.

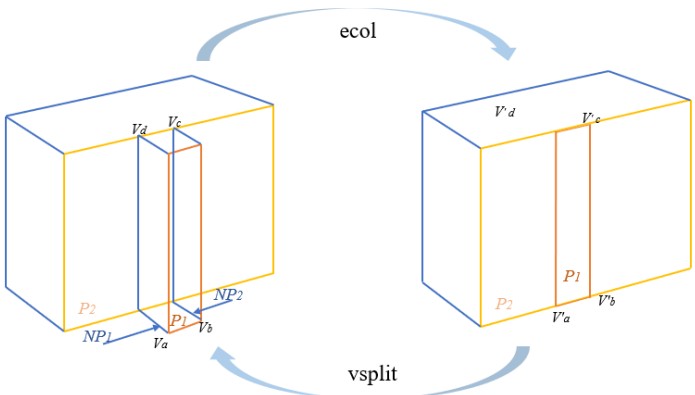

**Figure 4.** Detail plane simplification based on the facet shift algorithm.

In contrast, the process of reconstructing $M$ from $M_0$, that is, the reduction process (vsplit), is the inverse process of the simplified process, accomplished by adding the original nodes *Va*, *Vb*, *Vc*, and *Vd* and two noncharacteristic surfaces $NP_1$ and $NP_2$ to $M_0$.

The above process can be expressed as

the simplified process (ecol): $(M = M_n) \xrightarrow{ecol_{n-1}} M_{n-1} \cdots M_2 \xrightarrow{ecol_1} M_1 \xrightarrow{ecol_0} M_0$;

the reduction process (vsplit): $M_0 \xrightarrow{vsplit_0} M_1 \xrightarrow{vsplit_1} M_2 \cdots M_{n-1} \xrightarrow{vsplit_{n-1}} (M = M_n)$.

Therefore, the LOD transformation of a 3D building model can be composed of a detailed model $M$ and a set of information records of the plane translation process ($\{ecol_{n-1}, \ldots \ldots , ecol_0\}$), which can be expressed as $PM = \{M, \{ecol_{n-1}, \ldots \ldots , ecol_0\}\}$. In order to correctly restore the original detailed model's topological structure in the triangular mesh reconstruction of the 3D model, it is necessary to record the relevant information of each plane translation. In order to realize the continuous transformation of the level of detail of the 3D building model, it is necessary to improve the data structure of the 3D building model.

### 2.4. The Continuous LOD Topological Data Structure of A 3D Building Model

In order to reconstruct the triangular mesh of a 3D building model, it is necessary to record the relevant information of each translation surface. This paper presents a continuous LOD topological data structure based on improved geometric data organization (as shown in Figure 5), which includes geometric attribute data (vertices and triangular patches), topological relationship data (pointer pointing), and geometric data state (*bInUse*). The vertex is indexed by the ordinal number of the point in the vertex list, and the triangle is indexed by the ordinal number of the triangular patch in the triangular patch list. Each point or triangular patch has its fixed position in the data structure.

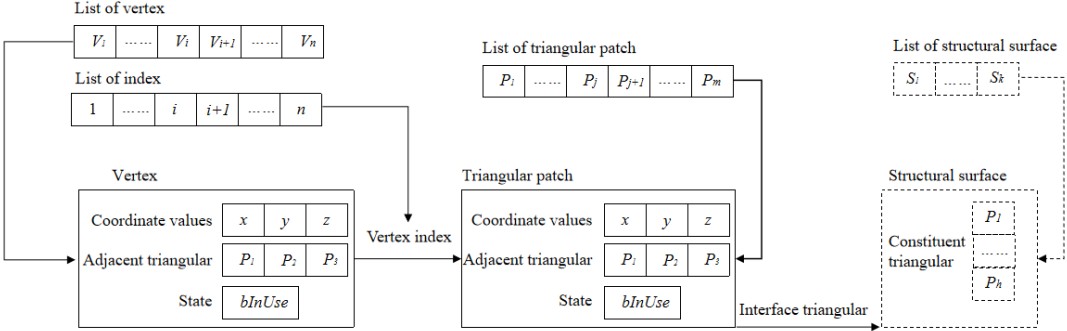

**Figure 5.** The continuous level-of-detail (LOD) topological data structure.

Geometric data include the geometric data carried by the original 3D building model and the new geometric data generated by spatial calculation in each stage of scale transformation. The geometric data in a state of being used indicate that they are being used to express surface or volume objects at some stage of scale transformation and need to be drawn in the visualization phase. Otherwise, they should not be drawn. By setting the characteristic surface of the model and vertices constituting the feature surface, the visibility of geometric data in the process of model scale transformation can be controlled, and 3D building models of different scales in continuous LOD can be represented.

When the secondary characteristic surface is merged into the dominating characteristic surface, the coordinate value of the prehidden vertex in the vertex list is preserved, the use state (*bInUse*) is changed into the nonuse state (**N**), while the coordinate value of the prehidden characteristic surface in the triangular patch list is preserved and the use state (*bInUse*) is changed to the nonuse state (**N**); in the meantime, the relevant data structure of the surfaces is updated.

In contrast, when the model details are restored, the root is used. According to the index number of additional vertices, the use state (*bInUse*) of the corresponding hidden points in the vertex list is changed into the use state (**Y**), while the use state (*bInUse*) of additional vertices is changed to the

nonuse state (**N**). In the meantime, according to the index number of additional characteristic surfaces, the use state (*bInUse*) of the corresponding hidden surfaces in the vertex list is changed into the use state (**Y**), and the use state of additional characteristic surfaces is changed to the use state (**Y**). State (*bInUse*) is a nonuse state (**N**). Simultaneously, the relevant facet data structure is updated.

To ensure the integrity of the recorded data, the relevant information of the translation plane must be able to directly or indirectly reflect three aspects of data:

1. The coordinate value of the prehidden vertex $N(v_j)$, and the index of the prehidden vertex;
2. Triangular patch $p_e$, $p_f$ that is connected to the prehidden vertex, and the indexes of the two triangular patches $N(p_e)$ and $N(p_f)$;
3. The indexes $(N(p_a), N(p_b), N(p_c)\ldots\ldots)$ of the noncharacteristic surface, which need to be updated.

The information records in the facet shift process are defined as follows:

$$R = \left\{v_j, N(v_j), p_e, N(p_e), p_f, N(p_f), N(p_a), N(p_b), N(p_c)\cdots\right\}. \tag{2}$$

The process of simplifying and restoring the details of 3D building models is actually the process of defining the data use status (*bInUse*) at specified locations in the data structure, while $N(v_j)$, $N(p_e)$, and $N(p_f)$ play the role of "location assignment", which is the key to ensuring the accurate process of simplifying and restoring the details of the whole model.

The content to be recorded in the facet shift process can be divided into two parts: vertex data and topological information. The vertex data represent the information of the prehidden points, and the topological information reflects the additional and updated characteristic patch information after the facet shift, both of which are persisted by the stack. In the process of continuous LOD model transformation, the earliest translated feature surface is restored at the latest point, and the last translated feature surface is restored at the earliest point. Considering the sequence of characteristic surface restoration in the process of continuous LOD model transformation, the stack which can process the messages in a first in, first out (FIFO) sequence is used to store the transformation information. The change-in and change-out operation of the transformational information stack is to move the specific pointer to change the LOD of the 3D building model. Figure 6 is an input–output operation diagram of the transformational information stack.

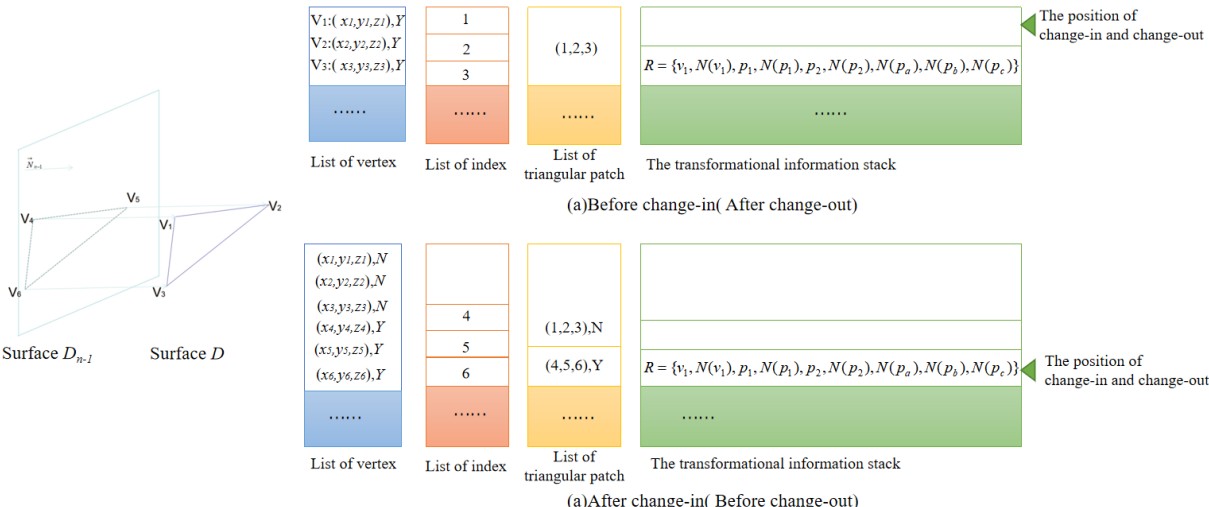

**Figure 6.** Change-in and change-out operation in a transformational information stack.

When changing in, the position pointer of the stack moves down, the change information record is pushed out, the additional vertices are added to the vertex list, and the corresponding indexes and surfaces are also updated in their list. When the 3D building model is refined and changed

out, the position pointer moves up, the change information record is pushed in, and the use state of additional vertices in the vertex list (*bInUse*) becomes the nonuse state (**N**). Furthermore, the use state (*bInUse*) of the corresponding indexes and the surfaces change into the nonuse state (**N**), and the 3D building model can be simplified. Simply put, by the change-in and change-out operation in the transformational information stack, models in each LOD can be obtained.

## 3. Continuous LOD Transformation Method for 3D Building Models

The LOD transformation of 3D building models includes two processes: simplification and restoration; each one is the other's inverse process. Taking the simplification process as an example, this includes wall, roof, and external appendage simplifications. Considering that the information characteristics of normal urban 3D building structures are mainly reflected in the walls and roofs, this work focused on the processing of the walls and roofs.

The simplification of concave–convex wall structures is to compare the two surfaces of a wall. The relative position of the two characteristic surfaces is determined by comparing the distance from the two characteristic surfaces to the center of the model. The translation direction of the plane is judged by the proportion of the two feature planes, so as to simplify the wall. All these simplification and restoration processes mainly use the facet shift algorithm, which is based on minimal features. Roof simplification first identifies the wall surface by prior knowledge (wall vertical ground), identifies the roof surface by the angle with the wall plane, compares the roof plane by two, and judges the translation direction of the plane by the area of the two characteristic planes, so as to simplify the wall surface. The flowchart of continuous LOD transformation for 3D building models is shown in Figure 7.

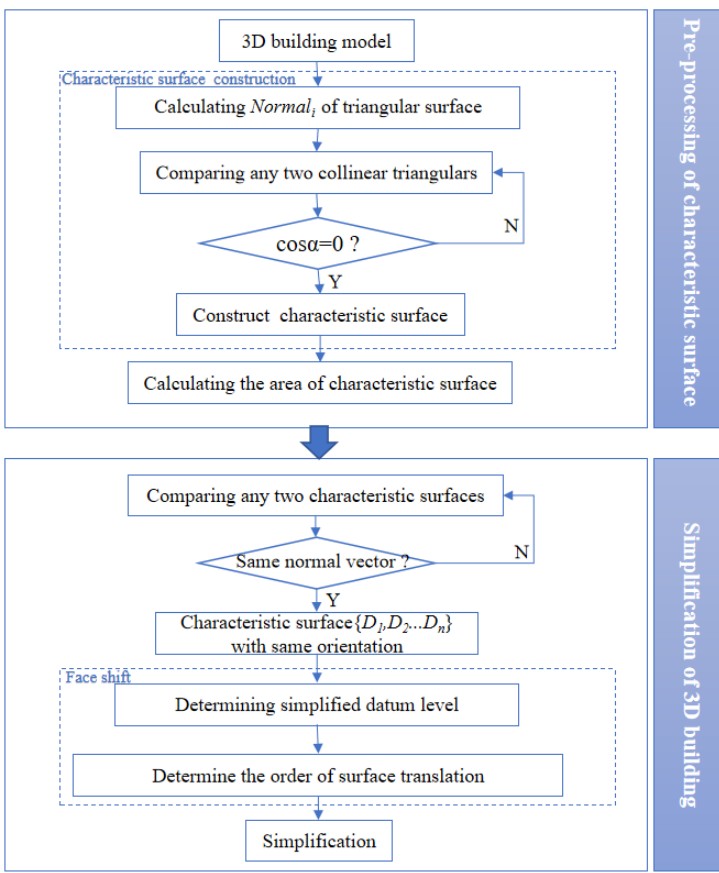

**Figure 7.** Flowchart of continuous simplification for 3D building models.

### 3.1. Pretreatment of Characteristic Surface

The 3D model is composed of triangular facets (Figure 8), while the LOD transformation of the 3D building model is based on the whole plane surface. Before the LOD transformation of the 3D building model, the triangular facets on the same plane should be merged if the normal vector of the triangular facets are the same and share a collinear point; then, the characteristic surface is constructed. The important value of each characteristic surface can be obtained by adding the area of the triangular facets from the same surface. The less important surface should always give up its site characteristics and inherit the site characteristics from surfaces with a higher importance level. The steps are as follows:

1. Calculating normal vectors:

$$Normal_i = (V_{i2} - V_{i1}) \times (V_{i3} - V_{i1}). \tag{3}$$

2. Determining coplanar triangles:

$$\cos \alpha = \frac{Normal_i \times Normal_j}{\left| Normal_i \times Normal_j \right|}. \tag{4}$$

If $\cos \alpha = 0$, then the two triangles are coplanar.

3. Calculating the area of each characteristic surface to determine the moving direction (the total area of the coplanar triangle):

$$S_\Delta = \sqrt{p \times (p-a) \times (p-b) \times (p-c)} \tag{5}$$

$$p = \frac{a+b+c}{2} \tag{6}$$

$$S_{surface} = \sum_{i=1}^{n} S_{\Delta i}. \tag{7}$$

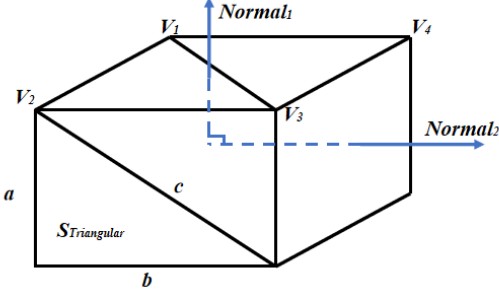

**Figure 8.** Triangular surface construction of the 3D model.

### 3.2. Simplification of the 3D Building Model

Here, the 3D building model mainly uses the facet shift algorithm based on minimal features and records the geometric data changes in the intermediate process of LOD transformation with the continuous LOD topological data structure. By storing the geometric point data and edge data information generated in the process of simplifying the datum level from outer surface $D_n$, then the usage status of geometric data generated can be changed in different stages, the characteristic surfaces can be restored at different levels, and, finally, the details of different stages can be restored without preparing the models of each LOD level in advance, thus realizing the continuous LOD transformation of the 3D building model. The steps of facet shift are as follows (Figure 9):

1.  Confirm the datum level $S_{target}$ where $S_{\blacktriangle max}$ is located.
2.  Calculate the distance $d_i$ from the center point of $S_{surfacei}$ to the datum level, and sort the planes $(D_1 < D_2 < \ldots \ldots D_n)$ according to the distance, from small to large.
3.  For each simplification step, only the outermost plane $D_n$ is simplified to the sub-outermost layer $D_{n-1}$.
4.  Until simplified to $S_{target}$.

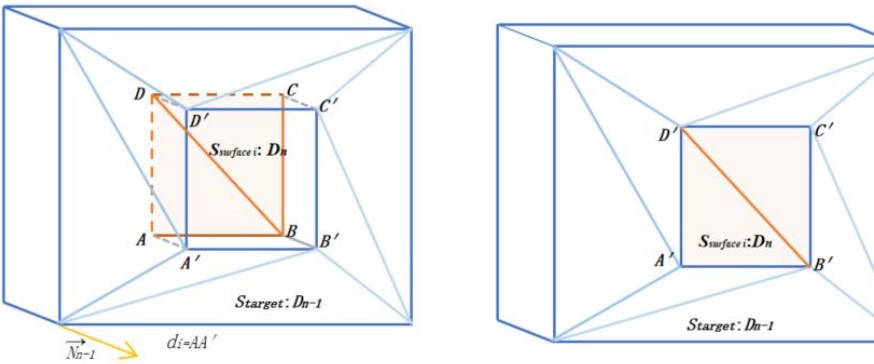

**Figure 9.** Model transformation based on continuous LOD topological data structure.

In the process of plane translation, the organization process of geometric data is as follows:

1.  Set point stack $ST_V$ and side stack $ST_E$ to store newly generated points and edges, respectively.
2.  New vertices A′, B′, C′, and D′ have their coordinate values set to be the projected values that vertices A, B, C, and D project along normal vector to Surface $D_{n-1}$.
3.  New edges A′B′, A′D′, B′C′, and D′C′, and set the default state of *bInUse* = **N**.
4.  Update the use stage of surface $D_n$ by replacing edges AB, AD, BC, and DC (setting edges AB, AD, BC, and DC as the nonuse state and A′B′, A′D′, B′C′, and D′C′ as the use state); then, the surface $D_n$ is simplified to surface $D_{n-1}$.

*3.3. General Transformation Method Compared with Improved Data Structure Transformation Method*

Figure 10 compares the transformation operating principles between the general model transformation method and the proposed model transformation method in an abstract way.

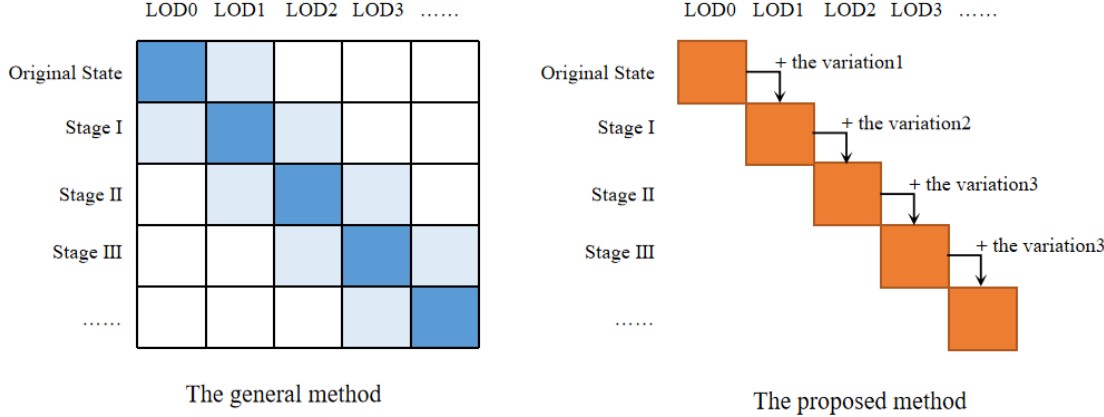

**Figure 10.** General transformation method compared with method based on continuous LOD topological data structure.

As for the general method, in the original state (the detail of the model was less than expected), models in LOD0 and LOD1 were loaded, the transparency of LOD1 model was set as 0, and the

transparency of the LOD1 model was set as 1. At this stage, the LOD1 model was actually loaded, where it was "invisible". From the original state to stage I (the detail of the model was more than expected than that in the original state), the transparency of the LOD1 model was changed from 0 to 1, while the transparency of the LOD0 model was changed from 1 to 0, with the LOD0 model fading and the LOD1 model rising, so that the transformation from the original state to stage I could be accomplished.

As for the proposed method, in the original state, models in LOD0 were the only loaded models. From the original state to stage I, the transformation from LOD0 to LOD1 was in real time, and the increased part was the increased details and the corresponding vertex and patches.

Above, part of the model reduction process was introduced through two methods, and both the rest of the part and the simplification process can also be done in the same manner.

## 4. Experiments and Analysis

In this study, a 3D building model with a 3DS data structure was used. There were two buildings in the model. The experimental data had obvious parallel structure characteristics in geometric form. On the platform of Visual Studio 2012, C++ language was used to read the point and surface information of the 3D model; then, OpenGL was used to process the model and display the 3D model.

The results of the algorithm test are shown in Figure 11. In this experiment, 4,991,123 triangular patches and 39,703 data points were processed. In the test process, the related hardware was an Intel Core i5-8250U processor and DDR4 240008GB memory. The test results showed that the 3D building model based on the data structure in this paper performed well in continuous LOD transformation. In the transformation process, the edges between triangles of different types of structures (mainly walls and roofs) were good, the picture was smooth, and the data consumed by the system during the running of the test model took up 22.68 M of memory. Figure 11 shows that in the process of transformation, the system consumed 22.68 M of memory. Under the condition of setting four levels of detail, the visualization results of each level of detail in the process of the 3D building model changed from the original state to stage III.

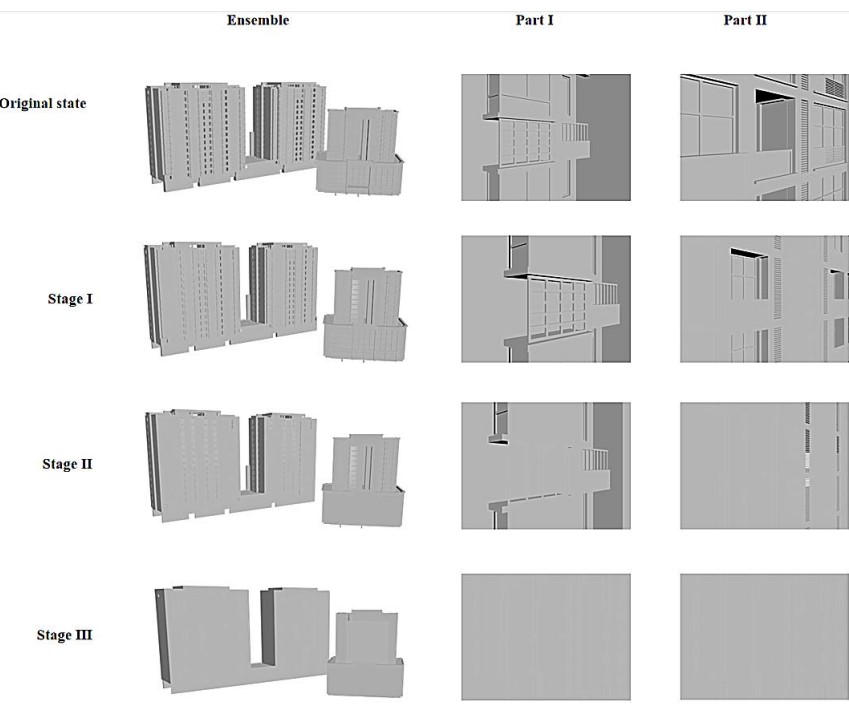

**Figure 11.** Continuous LOD transformation of 3D urban models (part).

The model had detailed architectural features such as window frames in the original state. In the process of simplification based on the data structure in this paper, the surface of the outermost parallel structure was projected to the inner plane to obtain the corresponding surface of the outermost parallel structure in the simplified state. At the same time, the outermost parallel structure was recorded and the *bInUse* was changed into **N**, which simplified the model and had information to restore it to the original state. In this way, the details of the model were gradually omitted until the model was simplified to stage III (at this point, the model could not be further simplified).

Table 1 shows the efficiency of the 3D model transformation while using the proposed method under experimental conditions (time-consuming error was 0.01 s).

**Table 1.** Efficiency statistics of 3D model transformation process.

|  | The Original State | Stage I | Stage II | Stage III |
|---|---|---|---|---|
| Total number of patches stored in the list of patches | 499,128 | 607,977 | 641,402 | 664,449 |
| Time taken for the simplification process from the original model to stage $i$ ($i$ = 1, 2, 3) (without rendering time) |  | 0.51 | 0.59 | 0.64 |
| Time taken for the reduction process from stage $i$ ($i$ = 1, 2, 3) to the original model (without rendering time) |  | 0.11 | 0.13 | 0.15 |

Figure 12 compares the efficiency variance between the general 3D building model continuous transformation method of fading and this method. The evaluation indexes are (1) memory consumption and (2) transformation time when loading and transforming. The index values were obtained by comparing the caching and background processing under the premise of consistent rendering stage. The general transformation method needs to pregenerate multiple LOD 3D building models, while the method in this paper synchronizes in the process of simplification. The variable intermediate quantities, which were directly used in the reduction process, were calculated and recorded. As can be seen from Figure 12, the general method loaded more facets and required more memory space in the transformation process. The more complex the loading model, the more facets that need to be processed in the transformation process, and the longer the transformation process takes, the more likely it is to cause the images to not be fluent.

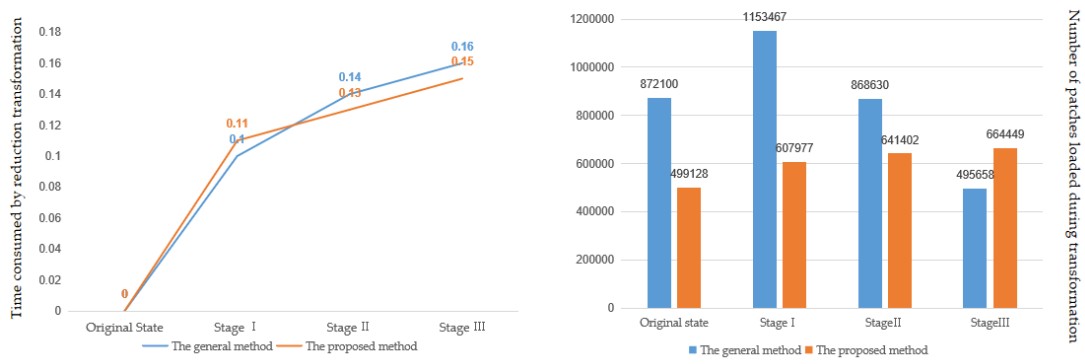

**Figure 12.** Comparison between the general method and the proposed method.

## 5. Conclusions

In this work, a data structure of continuous LOD transformation of parallel structure buildings was designed. By calculating and storing the process information of feature surfaces and points in the process of model transformation, the continuous LOD transformation of a 3D building model was realized by utilizing the process information. In the process of LOD transformation, the information of the translation process of the recording plane was set, and the position of each phase of the LOD transformation was determined by setting the state of the "new" surface generated by the same plane at different stages, thus affecting the details of the model. From the overall transformation effect, the data structure proposed in this paper can better meet the needs of smooth transformation

for 3D building models in different LODs. Compared with discrete LOD transformation, which needs to generate and store many simplified models in advance, the data structure proposed in this paper only stores all vertices in memory once in the process of LOD transformation. It solves the shortcoming of repeated storage of some vertices when discrete LOD stores multiple simplified models, occupies less storage space, and can better alleviate computer storage pressure. However, 3D building models generated through this method are limited in many applications. The simplified models may cause calculation errors in some application fields such as building area analysis, and for the reason that the topology constraints are considered, the real-time running efficiency of the method may be slightly slower when the continuous LOD scheme is adopted. Although the complexity of real 3D urban buildings is considered, it is still not comprehensive enough. Reasonably dealing with complex structure information in LOD transformation without affecting the visual effect will be a future research direction.

**Author Contributions:** Conceptualization, Siyi Li and Wenjing Li; Methodology, Siyi Li and Wenjing Li; Software, Shengjie Yi; Validation, Zhiyong Lin; Formal Analysis, Siyi Li; All authors contributed to the interpretation of the results and the writing of the paper.

**Funding:** This research was funded by the National Natural Science Foundation of China, grant number 41271449.

**Acknowledgments:** The authors would like to thank the editor and three anonymous reviewers for very thoughtful reviews on the previous version of this paper.

**Conflicts of Interest:** The authors declare no conflict of interest.

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
