# Peer review of "Method for 3D City Building Continuous Transformation Based on an Improved LOD Topological Data Structure"

_ijgi, doi:10.3390/ijgi8110504_

Round 1

Reviewer 1 Report

First sentence of the abstract contains repetition of ‘the granularity in different settings’

Abstract is not understandable. It contains of only 3 sentences, among which, one sentence is 5 lines long.

In order to be more understandable I suggest you to rewrite the abstract, correct errors (also typos) to emphasize the problem you are trying to resolve  and the goals of the paper

What is “jump” phenomenon?  Clarify

LOD – define

When you write about LOD you wrote in a context of a video game. Before this you should describe LOD technology in more detail and connect with why you mentioned video games

Generally, abstract and introduction should be rewritten in order to better describe the following text

Typos should be corrected.

Reviewer 2 Report

This paper is interesting because it aims to optimize the visual effects of continuous transformation of 3D building models in LOD. As an intuitive tool for describing cities, granularity in different settings is very important for granularity in different settings to improve the efficiency and reliability of city visualization. The great thing about this paper is that it demonstrates that this algorithm can support continuous LOD transformations with a good effect on 3D urban model patching with less storage capacity.

Based on the above, I would appreciate it if you could reconsider the following points.

・Purpose of the research results

Am I correct in understanding that this paper will create a model specialized for display optimization, apart from the original data? If so, it would be better to specify that it cannot be used for precise spatial analysis as a limiting condition.

・Review of previous studies

Throughout this paper, there is a lack of commonly cited major articles in the references to 3D building models that are presented. For example, it is recommended that the following major papers be added to the preceding research to reconfirm the history of the academic world.

An improved LOD specification for 3D building models

https://www.sciencedirect.com/science/article/pii/S0198971516300436

Allele-Sharing Models: LOD Scores and Accurate Linkage Tests

https://www.sciencedirect.com/science/article/pii/S0002929707602109

Model simplification using vertex - clustering

http://webdocs.cs.ualberta.ca/~anup/Courses/604_3DTV/Presentation_files/Polygon_Simplification/4-1.pdf

・About Figure 10

In the figure, please clearly show the difference between the static LOD and the continuous LOD. The main differences are hard to see in this figure.

・About Table 1

"Total number of patches stored in the list of patches" refers to Stage II (641402) and Stage III (664449).

On the other hand, "The proposed method" in Figure 11 is Stage II (664449) and Stage III (641402). Aren't these the same?

・About Figure 11

It is difficult to see the simplification status in stages I to III from the original model. Two enlarged views are shown in the original model, but the same enlarged views should be clearly shown in all stages I through III. I would also like to see a side-by-side diagram of the same model with static LOD. Without this, it will not be clear what the differences are.

・About Figure 12

For the number of patches in the Original state, the number 499128 in The general method and the number 872100 in The proposed method must match one or the other. Otherwise, the changes in the values of stages I to III cannot be compared.

Reviewer 3 Report

Line 14, duplicated semantics in the abstract: "the granularity in different settings it’s very important for the granularity in different settings to improve the efficiency..." Line 35, spelling mistake "witch". Please bring into correspondence with the figure citation. In the experimental analysis part, you evaluated the efficiency of your method - total number of stored patches, the time lost during the simplification process, and the time lost during the reduction process. But after that, you evaluated the performance of the traditional transformation method and the proposed one by comparing the number of loading patches and the time lost during the reduction process, please explain the reason why you use two inconsistent evaluation index systems. Line 286, what’s the purpose of calculating the total area of each characteristic surface? Line 357, you mentioned that "As can be seen from Fig. 12, the traditional method loads more facets and requires more memory in the transformation process". How to explain it? In figure 12, the picture on the right showed the better performance with the general method than the proposed method when comparing the number of patches loaded at stage Ⅲ, but it was ignored in the experiment analysis part, please describe and analyze this part. Since the data structure used in this paper is based on stack, there should be some limitations while operating the model detail transformation for the 3d buildings. It would be better to indicate the limitations while evaluating your method at the conclusion part or somewhere else appropriate.

Round 2

Reviewer 1 Report

Reads nicely now. Authors reconsidered all comments and included corrections in the paper.

Author Response

The manuscript has been polished for a better reading experience. 

Reviewer 2 Report

Thank you for your comments and corrections to the manuscript.
I am very satisfied and have no further comment.

Author Response

(The authors gave the same response as above.)

Reviewer 3 Report

The authors have answered all the questions and revised the paper.

Author Response

(The authors gave the same response as above.)
